# Fast Detection of 10 Cannabinoids by RP-HPLC-UV Method in *Cannabis sativa* L.

**DOI:** 10.3390/molecules24112113

**Published:** 2019-06-04

**Authors:** Mara Mandrioli, Matilde Tura, Stefano Scotti, Tullia Gallina Toschi

**Affiliations:** 1Department of Agricultural and Food Sciences, Alma Mater Studiorum-University of Bologna, Viale Fanin 40, 40127 Bologna, Italy; mara.mandrioli@unibo.it (M.M.); matilde.tura2@unibo.it (M.T.); 2Shimadzu Italia, Via G. B. Cassinis 7, 20139 Milano, Italy; sscotti@shimadzu.it

**Keywords:** cannabinoids, *Cannabis sativa* L., HPLC, validation

## Abstract

Cannabis has regained much attention as a result of updated legislation authorizing many different uses and can be classified on the basis of the content of tetrahydrocannabinol (THC), a psychotropic substance for which there are legal limitations in many countries. For this purpose, accurate qualitative and quantitative determination is essential. The relationship between THC and cannabidiol (CBD) is also significant as the latter substance is endowed with many specific and non-psychoactive proprieties. For these reasons, it becomes increasingly important and urgent to utilize fast, easy, validated, and harmonized procedures for determination of cannabinoids. The procedure described herein allows rapid determination of 10 cannabinoids from the inflorescences of *Cannabis sativa* L. by extraction with organic solvents. Separation and subsequent detection are by RP-HPLC-UV. Quantification is performed by an external standard method through the construction of calibration curves using pure standard chromatographic reference compounds. The main cannabinoids dosed (g/100 g) in actual samples were cannabidiolic acid (CBDA), CBD, and Δ9-THC (Sample L11 CBDA 0.88 ± 0.04, CBD 0.48 ± 0.02, Δ9-THC 0.06 ± 0.00; Sample L5 CBDA 0.93 ± 0.06, CBD 0.45 ± 0.03, Δ9-THC 0.06 ± 0.00). The present validated RP-HPLC-UV method allows determination of the main cannabinoids in *Cannabis sativa* L. inflorescences and appropriate legal classification as hemp or drug-type.

## 1. Introduction

*Cannabis* is classified into the family of Cannabaceae and initially encompassed three main species: *Cannabis sativa*, *Cannabis indica*, and *Cannabis ruderalis* [1]. Nowadays, Cannabis has only one species due to continuous crossbreeding of the three species to generate hybrids. In fact, all plants are categorized as belonging to *Cannabis sativa* and classified into chemotypes based on the concentration of the main cannabinoids. Depending on the THCA/CBDA ratio, some chemotypes have been distinguished. In particular, chemotype I or “drug-plants” have a TCHA/CBDA ratio >1.0, plants that exhibit an intermediate ratio are classified as chemotype II, chemotype III or “fiber-plants” have a THCA/CBDA ratio <1.0, plants that contain cannabigerolic acid (CBGA) as the main cannabinoid are classified as chemotype IV, and plants that contain almost no cannabinoids are classified as chemotype V [2,3,4,5].

Recently, in Italy the interest in *Cannabis sativa* L. has increased mainly due to the latest legislation (Legge n. 242 del 2 dicembre 2016) [6]. As a consequence, there is a request to develop cost-effective and easy-to-use quantitative and qualitative methods for analysis of cannabinoids.

The Italian regulatory framework has classified two types of *Cannabis sativa* L. depending on the content of Δ9-THC. In particular, fiber-type plants of *Cannabis sativa* L., also called “hemp”, are characterized by a low content of Δ9-THC (<0.2% *w/w*). If the content of Δ9-THC is >0.6% *w/w*, it is considered as drug-type, also called “therapeutic” or “marijuana”. 

Industrial hemp is used in several sectors, such as in the pharmaceutical, cosmetic, food, and textile industries, as well as in energy production and building. In general, fiber-type plants are less used in the pharmaceutical field, where drug-type plants are more often employed [5]. However, there is also an increased interest in hemp varieties containing non-psychoactive compounds. In fact, the European Union has approved 69 varieties of *Cannabis sativa* L. for commercial use [7].

Hemp has a complex chemical composition that includes terpenoids, sugars, alkaloids, stilbenoids, quinones, and the characteristic compounds of this plant, namely cannabinoids. *Cannabis sativa* L. has several chemotypes, each of which is characterized by a different qualitative and quantitative chemical profile [5]. The cannabinoids, terpenes, and phenolic compounds in hemp are formed through secondary metabolism [3,8]. The term “cannabinoid” indicates terpenophenols derived from *Cannabis*. More than 90 cannabinoids are known, and some are derived from breakdown reactions [8]. Gaoni and Mechoulam [9] were the first to define cannabinoids “as a group of C_21_ compounds typical of and present in *Cannabis sativa*, their carboxylic acids, analogs, and transformation products”. Currently, cannabinoids have been classified according to their chemical structure, mainly seven types of cannabigerol (CBG); five types of cannabichromene (CBC); seven types of cannabidiol (CBD); the main psychoactive cannabinoid Δ9-tetrahydrocannabinol (Δ9-THC) in nine different forms including its acid precursor (Δ9-tetrahydrocannabinolic acid, Δ9-THCA); Δ8-tetrahydrocannabinol (Δ8-THC), which is a more stable isomer of Δ9-THC but 20% less active; three types of cannabicyclol (CBL); five different forms of cannabielsoin (CBE); seven types of Cannabinol (CBN), which is the oxidation artifact of Δ9-THC; cannabitriol (CBT); cannabivarin (CBDV); and tetrahydrocannabivarin (THCV) [10,11]. THC, CBD, CBG, CBN, and CBC are not biosynthesized in *Cannabis sativa*, and the plant produces the carboxylic acid forms of these cannabinoids (THCA, CBDA, CBGA, CBNA, and CBCA). Cannabinoid acids undergo a chemical decarboxylation reaction triggered by different factors, mainly temperature. This decarboxylation reaction leads to the formation of the respective neutral cannabinoids (THC, CBD, CBG, CBN, and CBC) [12,13].

There are several methods to quantify cannabinoids [14,15,16,17,18,19,20,21], some of which require expensive mass spectrometry detectors [22,23,24,25]. Furthermore, there is a great deal of uncertainty around the use of gas chromatography (GC) for the titration of cannabinoids due to the high temperature of the injector and detector that can lead to the decarboxylation of cannabinoid acids if not derivatized correctly [26]. Moreover, recent studies have reported that cannabinoid acid decarboxylation is only partial, and as result the actual value is underestimated. An HPLC system allows for determination of the actual cannabinoid composition, both neutral and acid forms, without the necessity of the derivatization step [13].

It is necessary, in addition to honed methods, to develop new procedures with a view to discriminate different *Cannabis* varieties in order to identify and titrate cannabinoids in a simple way. These methods should ideally be fast, easy, robust, and cost-efficient as they can be used not only by research laboratories but also by small companies with a view on quality control. 

This study focuses on the development, validation, and step-by-step explanation of a rapid and simple HPLC-UV method for identification and quantification of the main cannabinoids in hemp inflorescences that can be easily reproduced and applied. The method described is focused on the quantification of CBD but can also be applied to check the levels of THC.

## 2. Results and Discussion

### 2.1. Method Development

The aim of this work was to develop a new analytical method for determination of the main cannabinoids in hemp samples. In fact, the method described below can be used as a routine quality control procedure and can be applied by the pharmaceutical industry, small laboratories, or even small pharmacies.

A crucial aspect for accurate identification and quantification of analytes is optimization of separation conditions, and therefore various preliminary tests were carried out (e.g., mobile phase, detection wavelength). Different mobile phases were tested, and trials were performed with different compositions and gradient elution to optimize the separation of all 10 target compounds considered (Appendix A). The greatest difficulty was that of separating CBD and THCV, which in many cases co-eluted. It was also difficult to separate the isomers Δ9-THC and Δ8-THC. The best resolution of cannabinoids was obtained using a chromatographic column and, as an eluent mixture, water with 0.085% phosphoric acid and acetonitrile with 0.085% phosphoric acid. 

The quantification of cannabinoids was made at 220 nm after testing different wavelengths (Appendix A). This wavelength represents the best compromise for all the cannabinoids considered and was selected to detect and integrate all compounds of interest within the dedicated concentration range. As far as chromatographic analysis is concerned, before using the instrument, the system was conditioned for 20 min by fluxing the eluent mixture in the instrument under the same conditions as the method, and then a chromatographic run was performed by injecting 5 μL of acetonitrile to verify that the chromatographic system was adequately cleaned. Simultaneously with the analysis of the sample, standard solutions were injected at different concentrations for the construction of calibration curves and to evaluate the separation and identification of each compound. The identification of cannabinoids was performed by comparing their retention times with those obtained by the injection of pure standards and by an enhancing procedure. Figure 1 shows a chromatogram of a standard mixture of cannabinoids and Figure 2 shows a chromatogram of a sample of hemp. 

Cannabinoids in different varieties of *Cannabis sativa* L. can be present in very different concentrations. In order to obtain good chromatographic separation and correct quantification, it may be necessary to dilute or concentrate the extract, performing two different injections. For example, in the case of high levels of CBDA or CBD it will be necessary to dilute the extract. For THC, it is often found at low concentration in hemp inflorescences, so it may be necessary to concentrate the extract before injection. In our case, 2 mL of filtered extract was dried using a weak nitrogen flow, and the dry extract was recovered in 500 μL of acetonitrile. 

### 2.2. Validation

#### 2.2.1. Precision

The precision of the method was measured by the expression of repeatability (*r*) and reproducibility (*R*). Precision was expressed through coefficient of variation (CV%).

#### 2.2.2. Repeatability, R

Table 1 shows data on the intraday and interday repeatability, evaluated as reported in Section 3.6, which demonstrates very high repeatability. In fact, the relative standard deviation (RSD) varied from 2.59 to 5.65 for intraday repeatability and from 2.83 to 5.05 for interday repeatability. In both cases, the highest RSD was found for CBDA, which is probably due its higher concentration compared to the other cannabinoids. 

#### 2.2.3. Reproducibility, R

The RSDs obtained in the reproducibility studies are shown in Table 1. The maximum RSD value was 2.13 for CBGA. The other cannabinoids show RSD values lower than 1.91, and the lowest of the RSDs was 0.09 for CBDA, which is probably due to the higher concentration of this cannabinoid.

#### 2.2.4. Recovery

The tests were performed by using three different concentrations to test the recovery values in the linearity range of the method.

Quantities of CBD (4, 8, and 24 μg/mL) were added, thus assessing concentrations similar to, higher, and lower than those found in samples.

Recovery was determined according to this modality for CBD and was 84.92%.

An evaluation of recovery on all the compounds present in the sample was carried out by proceeding with a further extraction with 10 mL of methanol-chloroform on the sample residue after the usual extraction; in this extract, some cannabinoids were present, and indirectly the percentage of recovery was determined. 

The percentage of recovery values, as shown in Table 1, were higher than 84.92% and can be considered very satisfactory. In fact, considering CBD, the percentages are higher than those previously reported in the literature [5].

#### 2.2.5. Detection Limit, LOD

The instrumental limit of detection was determined by the calibration curve, according to the formulas expressed in Section 3.6. The instrumental limit of detection (LOD) values obtained for CBDA and CBGA (Table 1) were lower, while those of CBG and CBD were comparable with similar methods described in literature [5,27]. Low LOD values were found also for the other cannabinoids (THCV, CBN, Δ-9 THC, Δ-8 THC, CBC, THCA), indicating that the method is sensitive.

#### 2.2.6. Quantification Limit, LOQ

The instrumental limit of quantification was determined by a calibration curve, according to the formulas expressed in Section 3.6, considering that the signal-to-noise method is particularly useful to quantify the cannabinoids present at lower concentrations, such as THC. As reported for the LODs, the instrumental limit of quantification (LOQ) values obtained for CBDA and CBGA (Table 1) were also lower than those reported in the literature, while those for CBG and CBD were comparable with those of other methods described for similar procedures [5,27]. In addition, the other cannabinoids (THCV, CBN, Δ-9 THC, Δ-8 THC, CBC, THCA) showed low LOQs. The instrumental noise was registered in µV, by performing 3 blank injections with the ASTM method [28] given by the instrument, and a maximum CV% of 3.49% was calculated for all individual compounds to determine the single LOD and LOQ, which was considered acceptable. 

#### 2.2.7. Linearity

In order to evaluate the linearity of the method, eight different points of standard mixture solutions were analyzed in triplicate by HPLC-UV.

The following equations are related to the calibration curves in a concentration range between 0.01–100 μg/mL: CBDA, y = 18955x − 1612.6 (*r*^2^ = 0.9999); CBGA, y = 19796x − 3475.7 (*r*^2^ = 0.9999); CBG, y = 18094x − 9195.3 (*r*^2^ = 0.9995); CBD, y = 13703x − 6009.5 (*r*^2^ = 0.9995); THCV, y = 18534x − 15213 (*r*^2^ = 0.9989); CBN, y = 34148x − 7943.1 (*r*^2^ = 0.9999); ∆9 − THC, y = 19893x − 31896 (*r*^2^ = 0.9981); ∆8-THC, y = 17526x − 18267 (*r*^2^ = 0.9987); CBC, y = 18590x − 4777.1 (*r*^2^ = 0.9999); THCA, y = 18239x − 8969.3 (*r*^2^ = 0.9998) (Table 1).

With the aid of the equation obtained from the calibration curve, the quantity of each cannabinoid was calculated.

To express the data relative to the content of the individual cannabinoid as a percentage (%, p/p) referred to the dried material, it is necessary to refer to the weight of the sample considering the dilution factor. The linearity in the concentration range analyzed was good for cannabinoid standards, being *r*^2^ > 0.998, as reported before. 

### 2.3. Cannabinoids in Hemp Samples

The method developed in this study was applied to quali-quantitative analysis of main cannabinoids in two samples of hemp inflorescences. The samples analyzed, belonging to the same variety of *Cannabis sativa* L., did not show a significant difference in the concentration of the target compounds. As shown in Table 2, CBDA is the only cannabinoid for which a different concentration was determined. The other cannabinoids had a similar or the same concentration (e.g., CBGA, CBG, CBN, Δ-9-THC, and Δ-8-THC) in both samples. THCV was not found in the hemp inflorescence samples analyzed, as shown in Figure 2 and Table 2. Δ-9-THC and Δ-8-THC were found at a low concentration, below the legal limit. Under the current legislation regarding *Cannabis sativa* L. cultivation [6,29], in fact, the total content of THC must not be higher than 0.2% and in any case within 0.6%. Indeed, only the hemp varieties reported in the *Common catalogue of varieties of agricultural plant species* can be cultivated without authorization [6,7]. These kinds of results confirmed that the analyzed samples were correctly classified as hemp, since the quantity of Δ8-THC and Δ9-THC was found to be lower than the limits established by the legislation. According to what is indicated in literature [30], in the hemp variety considered (Futura 75), the most present compound was CBDA, followed by CBD; all the other compounds were in very low amounts ranging from 0.01 to 0.06%. CBGA is the compound from which all other cannabinoids are biosynthesized [5], which is probably why it was found at a low concentration in both samples examined.

The number of cannabinoids in hemp samples is reported in Table 2.

## 3. Materials and Methods 

### 3.1. Chemicals, Standards and Apparatus

All chemicals used were of analytical grade. Methanol p.a CAS 67-56-1, chloroform p.a CAS 67-66-3, acetonitrile CAS 75-05-8, water CAS 7732-18-5, and orthophosphoric acid CAS 7664-38-2 were purchased from Sigma-Aldrich (St. Louis, MO, USA). Nitrogen, pure gas for analysis CAS 7727-37-9 was purchased from SIAD Spa (Bergamo, Italy). Standard mixture of phytocannabinoids 0.1% in acetonitrile: Cannabidiolic acid (0.01%) CAS 1244-58-2, cannabigerolic acid (0.01%) CAS 25555-57-1, cannabigerol (0.01%) CAS 25654-31-3, cannabidiol (0.01%) CAS 13956-29-1, tetrahydrocannabivarin (0.01%) CAS 31262-37-0, cannabinol (0.01%) CAS 521-35-7, tetrahydrocannabinolic acid (0.01%) CAS 23978-85-0, Δ-9-tetrahydrocannabinol (0.01%) CAS 1972-08-3, Δ-8-tetrahydrocannabinol (0.01%) CAS 5957-75-5, cannabichromene (0.01%) CAS Number 20675-51-8, were purchased from Cayman Chemical Company, (Ann Arbor, MI, USA). Cannabidiol 1.0 mg/mL in methanol CAS 13956-29-1: LGC Standards S.r.l., (Milan, Italy).

Analytical mill, IKA A11 Basic (IKA^®^ Werke GMBH & Co. KG, Germany). Analytical balance with precision of 0.1 mg, mod. E42, (Gibertini, Italy). Vortex vibrating shaker, mod. ST5, (Janke & Kunkel, Germania). Centrifuge mod. ALC, PK 120 (Thermo Electron Corporation, Massachusetts, USA). Termoblock heating block, mod. A120, (Falc, Italy). Natural ventilation stove. Sieve with 1 mm meshes. Tilting shaker. Ultrasound bath Branson 2150, (Danbury-CT, USA). Volumetric flasks of 1, 2, 10 and 25 mL. SOVIREL-type tubes with screw cap. Glass syringes with luer lock attachment, 0.45 μm nylon membrane filters. Microsyringes from 1 to 1000 μL. HPLC Cannabis Analyzer for Potency Prominence-i LC-2030C equipped with a reverse phase C18 column, Nex-Leaf CBX Potency 150 × 4.6 mm, 2.7 μm with a guard column Nex-Leaf CBX 5 × 4.6 mm, 2.7, UV detector and acquisition software LabSolutions version 5.84 (Shimazu, Kyoto, Japan).

### 3.2. Sampling

The samples were supplied by a company that produces industrial hemp. In particular, two samples (L11 and L5) of inflorescences of *Cannabis sativa* L. Futura 75 were analyzed, having come from the same land and harvested in August 2017, and supplied by Enecta Srl. Sampling of material was carried out on a population of hemp plants, according to a systematic path, so that the sample taken was representative of the particle, excluding the edges, taking the upper third of the selected plant as indicated in Reg. (EU) No 1155/2017 [31]. The sample was dried in an oven at 35 °C ± 1 to constant weight, and gross wood parts and seeds with a length of more than 2 mm were removed. The samples were then subjected to grinding and subsequent sieving through a sieve with 1 mm meshes. The sieved material was transferred into polypropylene containers and stored under nitrogen atmosphere, protected from light at a temperature of −20 °C until extraction. Three independent replicates were performed for each sample, and three HPLC injections were performed for each replication.

### 3.3. Cannabinoid Extraction

To extract cannabinoids, an aliquot of powder sample, about 25 mg, was weighed using an analytical balance; 10 mL of methanol-chloroform extraction solvent 9:1 (*v*/*v*) was added as reported by De Backer et al. (2009) [32], Jin et al. (2017) [33], and was placed first for 10 min on an oscillating oscillator set at 350 oscillations per minute and then for 10 min in an ultrasonic bath. The sample was centrifuged for 10 min at 1125 g, and the supernatant was removed. The extraction was performed twice. The two fractions containing cannabinoids were collected in a 25 mL volumetric flask and were brought to volume with methanol/chloroform (9:1, *v*/*v*). The samples were filtered with a 45 μm nylon filter. Two mL of the filtered extract was transferred to a glass tube. The solvent was removed, leading to dryness with the help of a weak nitrogen flow, and recovered with 500 μL acetonitrile. The solution was injected into an HPLC-UV.

### 3.4. Preparation of Standard Solution

Appropriate aliquots of a standard mixture of cannabinoids are diluted with acetonitrile to obtain solutions of known concentration, in particular eight points in a concentration range between 0.05 and 100 μg/mL (0.05, 0.50, 4.17, 8.33, 16.70, 25.00, 50.00, 100.00 μg/mL). The standard solutions were prepared to construct calibration curves for the 10 cannabinoids considered: CBDA, CBGA, CBG, CBD, THCV, CBN, Δ9-THC, Δ8-THC, CBC, and THCA. The standard solutions were stored away from light at a temperature of −20 °C. The stability of standard solutions stored at −20 °C was evaluated every week for 3 months with the HPLC-UV system, and no degradation of cannabinoids was found.

### 3.5. HPLC Conditions

For the RP-HPLC analysis, the column was thermostated at 35 °C, and the autosampler was thermostated to 4 °C. Sample concentration was 4 mg/mL, and injection volume was 5.0 μL. UV detection was used at 220 nm, and gradient elution was used at flow rate of 1.6 mL/min according to the following procedure. Eluent mixture: Water + 0.085% phosphoric acid (A), acetonitrile + 0.085% phosphoric acid (B). Gradient elution: 70% of B up to 3 min, 85% of B to 7 min, 95% of B to 7.01 up to 8.00 min, and 70% of B up to 10 min. The eluent mixture was previously filtered with a Millipore system equipped with a 0.2 μm nylon filter.

### 3.6. Validation Parameters

#### 3.6.1. Precision

Precision is the closeness of agreement among independent test results, obtained with stipulated conditions and usually in terms of standard deviation or relative standard deviation [34]. 

Precision was calculated with the following formula: CV% = [(SD/$\bar{x}$) × 100], where SD is the estimate of the standard deviation and $\bar{x}$ is the average of the replications made.

#### 3.6.2. Repeatability, R

The repeatability (intraday) of the method was evaluated by analyzing three replicates of the same sample, injected three times on the same day, performed by the same operator with the same method and instrument. The result corresponds to the arithmetic mean of the three determinations made considering the estimate of the standard deviation (SD) calculated on the three replicates performed. 

The repeatability (interday) of the method was evaluated by performing three replicates of the same sample, injected three times on three different days, performed by the same operator with the same method and instrument. The result corresponds to the arithmetic mean of the three determinations made considering the estimate of the standard deviation (SD) calculated on the three replicates performed.

#### 3.6.3. Reproducibility, R

Reproducibility was evaluated by the agreement between the results obtained on the same sample with the same procedure carried out by different operators in the laboratory and was measured with the coefficient of variation. 

#### 3.6.4. Recovery

Recovery is the fraction of analyte that was added to the sample being tested. Recovery was expressed as a percentage (R (%)) according to the following formula: R (%) = [(Cf − C)/Cc] × 100, where Cf is the endogenous amount of the cannabinoid in the sample plus the amount of standard added to the analyte under examination. C is the endogenous amount present in the sample not added with the standard. Cc is the amount of the standard analyte added to the sample.

#### 3.6.5. Detection Limit, LOD

The detection limit is the smallest amount or concentration of analyte in the sample that can be reliably distinguished from zero [34]. It can be calculated using the following formula: LOD = (3.3 × σ)/m, where: σ represents the residual standard deviation of the calibration curve and m represents the slope of the calibration curve. 

Furthermore, the LOD of the method from the signal (S)/noise (N) ratio can be determined as LOD: S/N = 3.

#### 3.6.6. Quantification limit, LOQ

The quantification limit is the concentration of analyte below which it is determinable with a level of precision that is too low with inaccurate results. The LOQ can be determined according to the following formula: LOQ = (10 × σ)/m, where σ represents the residual standard deviation of the calibration curve and m represents the slope of the calibration curve.

The LOQ of the method can also be determined by the signal-to-noise ratio (S/N): LOQ: S/N = 10.

#### 3.6.7. Linearity

Linearity can be tested by examination of a plot of residuals produced by linear regression of the responses on the concentrations in an appropriate calibration set [34].

In order to quantify the analytes of interest, the equation of the calibration curve obtained for each standard is used. The equation is: y = ax + b, where y = area of the analyte obtained by HPLC/UV analysis, a = slope of the calibration curve, x = unknown concentration (μg/mL) of analyte in the sample, b = intercept of the calibration curve.

## 4. Conclusions

One of the most relevant problems in analytical determinations for quality control, especially when there are legal problems related with quantitation, such as for cannabis, relates to the proficiency of laboratories. Therefore, detailed and validated procedures that are freely available are essential for the full understanding of any analytical step and its careful application. This is also true for “daily” methods that can be easily applied for quality control, carried out using traditional RP-HPLC and UV-Vis detectors, with less efficient performance than diode-array detectors but with lower costs, rendering them affordable even for small laboratories. 

The validated method described herein allows the quantitative determination of the 10 most relevant cannabinoids using a single wavelength (220 nm) in 8 min. A full separation is obtained, even in the elution sequence of a difficult resolution, of the group of peaks related to CBGA, CBG, CBD, and THCV (from 3.5 to 4.5 min).

The method is applied to cannabis inflorescences and involves extraction in methanol/ chloroform, drying of the extract, taking it up in acetonitrile and injection into an HPLC. The method has sensitivity and accuracy to discriminate samples with amounts of Δ-9- and Δ-8-THC (total THC content) that are below the limit of 0.2% from those that are subjected to legal restrictions in many EU countries, with a total THC content above 0.6%, which cannot be classified as hemp. Due to its simplicity and rapidity, it can be used to check raw material or crops during the harvesting period.

A detailed standard operating procedure (SOP), as a supplementary information file, is also available, so that any operator with basic knowledge of HPLC can easily apply it and make all the elution and calibration control checks using commercially available mixtures of standards, which are more affordable and sustainable than single cannabinoid standards in terms of costs and solvents used for calibration.

## Figures and Tables

**Figure 1 molecules-24-02113-f001:**
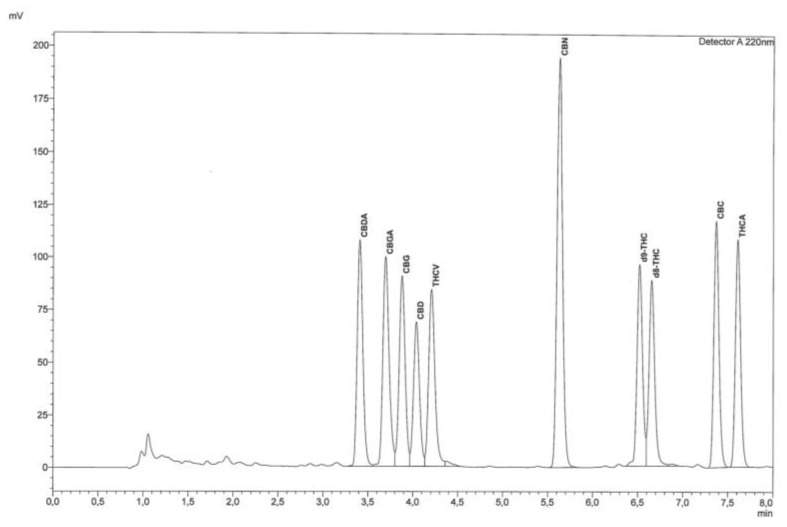
Chromatographic trace of a standard cannabinoid mixture analyzed by RP-HPLC-UV equipped with reverse phase C18 column.

**Figure 2 molecules-24-02113-f002:**
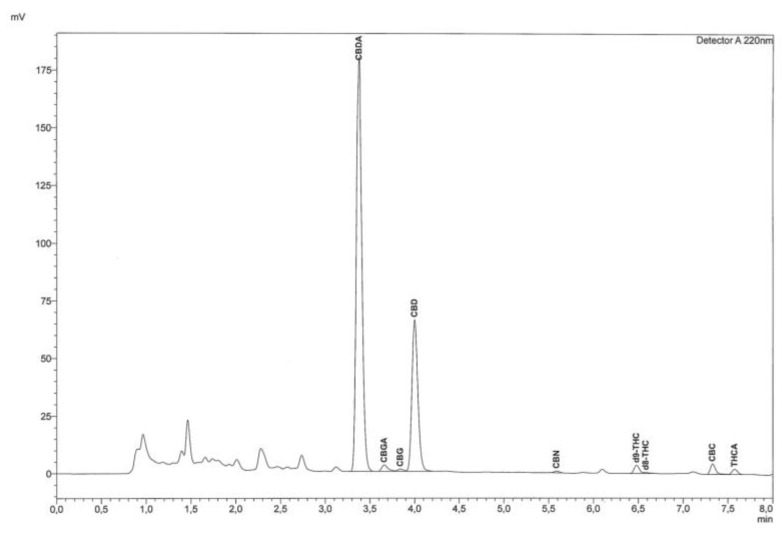
Chromatographic trace of *Cannabis sativa* L. inflorescence extract analyzed by RP-HPLC-UV equipped with a reverse phase C18 column.

**Table 1 molecules-24-02113-t001:** Validation parameters of RP-HPLC-UV method.

Compound	R^2^	^1^ LOD (µg/mL)	^2^ LOQ (µg/mL)	^3^ LOD (µg/mL)	^4^ LOQ (µg/mL)	Intraday (Repeatability) RSD	Interday (Repeatability) RSD	Reproducibility RSD	Recovery (%)
CBDA	0.9999	0.34	1.05	0.11	0.37	5.65	5.05	0.09	96.06
CBGA	0.9999	0.32	0.98	0.12	0.40	4.71	4.34	2.13	93.90
CBG	0.9995	0.62	1.87	0.13	0.45	3.34	2.83	0.91	94.60
CBD	0.9995	0.63	1.91	0.17	0.58	4.89	4.44	0.70	84.92
THCV	0.9989	0.95	2.87	0.15	0.49	-	-	N.d. *	N.d. *
CBN	0.9999	0.28	0.84	0.06	0.21	2.59	2.95	0.81	97.08
Δ9-THC	0.9981	1.25	3.79	0.15	0.50	3.05	3.22	0.13	99.69
Δ8-THC	0.9987	1.02	3.10	0.17	0.56	3.81	3.64	0.74	100
CBC	0.9999	0.29	0.88	0.11	0.36	5.3	4.78	0.89	98.68
THCA	0.9998	0.43	1.29	0.11	0.37	5.55	5.01	1.91	95.27

^1^ Limit of detection (LOD) determined by the calibration curves (Instrumental LOD = (3.3 × σ)/m). ^2^ Limit of quantification (LOQ) determined by the calibration curves (Instrumental LOQ = (10 × σ)/m). ^3^ LOD determined by the signal-to-noise ratio (Instrumental LOD: S/N = 3). ^4^ LOQ determined by the signal-to-noise ratio (Instrumental LOQ: S/N = 10). * Not detectable.

**Table 2 molecules-24-02113-t002:** Number of cannabinoids in hemp samples.

Cannabinoids										
Sample	CBDA (%)	CBGA (%)	CBG (%)	CBD (%)	THCV (%)	CBN (%)	Δ9-THC (%)	Δ8-THC (%)	CBC (%)	THCA (%)
*L11* *CV%*	0.88 ± 0.04 5.05	0.02 ± 0.00 4.34	0.02 ± 0.00 2.83	0.48 ± 0.02 4.44	N.d. *	0.01 ± 0.00 2.95	0.06 ± 0.00 3.22	0.03 ± 0.00 3.64	0.03 ± 0.00 4.78	0.03 ± 0.00 5.10
*L5* *CV%*	0.93 ± 0.06 6.48	0.02 ± 0.00 1.28	0.02 ± 0.00 1.73	0.45 ± 0.03 6.28	N.d. *	0.01 ± 0.00 1.49	0.06 ± 0.00 0.21	0.03 ± 0.00 2.20	0.02 ± 0.00 2.98	0.04 ± 0.00 7.17

* Not detectable.

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
