# Peer review of "Fast Detection of 10 Cannabinoids by RP-HPLC-UV Method in Cannabis sativa L."

_molecules, 2019, doi:10.3390/molecules24112113_

Round 1
Reviewer 1 Report
The manuscript numbered molecules-513726 deals with the fast analysis of 10 cannabinoids (the chromatographic run only lasts 8 min) in Cannabis Sativa L. by means of RP-HPLC-UV after a simple extraction with a mixture of methanol-chloroform 9:1, v/v.
The manuscript is well-written and organized. It also clearly explains background in the Introduction, method development and results. The authors offers a fast and cheap method as an alternative approach to more expensive LC-MS-based methods. In my opinion, the article only needs minor revision before publication on Molecules. Below, there are the points to be corrected:
- line 128: Please, change “accuracy” with “precision”. Repeatability and reproducibility are components of precision not accuracy. According to this, also modify the other parts of the manuscript (ex. Line 142).
- line 132: “The repeatability (intraday)…….”
- line 137: “The repeatability (interday)…..”
- lines 159-161: Please, explain clearly this part. It is not clear what Cf and Cc are. Is Cf the pre-extraction spike level and Cc the post-extraction spike level? C can simply be indicated as the analyte endogenous amount. You should also explain the criterion you used to choose the spike level: for example, two times the endogenous concentration.
- lines 169 and 179: Please specify hereand in Table 1 that these are instrumental LODs and LOQs since the calibration curves are built in solvent not in matrix.
Author Response
Response to Reviewer 1 Comments
Point 1: Line 128: Please, change “accuracy” with “precision”. Repeatability and reproducibility are components of precision not accuracy. According to this, also modify the other parts of the manuscript (ex. Line 142).
Response 1: We thank the reviewer for the correction, the suggested changes were made in the revised version.
Point 2:line 132: “The repeatability (intraday)…….”
line 137: “The repeatability (interday)…..”
Response 2:We would like to thank the reviewer for the correction, the suggested changes were made in the revised version.
Point 3: Lines 159-161: Please, explain clearly this part. It is not clear what Cf and Cc are. Is Cf the pre-extraction spike level and Cc the post-extraction spike level? C can simply be indicated as the analyte endogenous amount. You should also explain the criterion you used to choose the spike level: for example, two times the endogenous concentration.
Response 3: We thank the reviewer for the suggestion, as indicated clarification was provided in the revised manuscript. As suggested, the enrichment made for the recovery calculation was better specified.
Point 4: Lines 169 and 179: Please specify here and in Table 1 that these are instrumental LODs and LOQs since the calibration curves are built in solvent not in matrix.
Response 4: This specification and more detail were included, as suggested by the reviewer.
Thanking you in advance for your kind attention,
Sincerely,
Tullia Gallina Toschi
Reviewer 2 Report
Summarized comments:
* The subject falls within the general scope of the journal.
* The manuscript gives a new contribution a HPLC method
* Interpretations and conclusions are not completely satisfactory
General comments to the manuscript:
This study may be a valid piece of science. The clarity of the presentation of the results is satisfactory but it needs to be re-worked by evidencing some declaration that are not fully investigated inside the experimental data discussions.
Line 295-296: “method described is very fast and simple, robust, sensitive, repeatable, 295 reproducible and cost-efficient”. The declaration looks like a comparison with other methods that it is not the main focus of the study.
Formally the comparison with other method does not have any scientific dissertation inside the paper, moreover the authors repeat in the conclusion some concept that were describe in the introduction focusing their attention on derivatization procedures that are in use in the GC-MS analytical procedures.
The main problem is that the Authors focused their attention only on several compounds. Therefore it is not convenient that they stress this point.
Other point is that they did not declare how much their method is more sensitive than the other, they only describe the analytical procedures.
About the analytical procedures they declare that were conducted by using:
Line 126 “in-house validation”;
That is not scientifically acceptable and the major point is about the use of an internal standard in order to normalize the instrumental error procedures witch depends from the instrumental noise that could be change during the time needed to perform all the experiments.
Author Response
Response to Reviewer 2 Comments
Point 1:Line 295-296: “method described is very fast and simple, robust, sensitive, repeatable, 295 reproducible and cost-efficient”. The declaration looks like a comparison with other methods that it is not the main focus of the study.
Response 1: We thank the reviewer for the suggestion, we eliminated the sentence here, reporting part of the content in the discussion and the conclusions have been revised.
Point 2: Formally the comparison with other method does not have any scientific dissertation inside the paper, moreover the authors repeat in the conclusion some concept that were describe in the introduction focusing their attention on derivatization procedures that are in use in the GC-MS analytical procedures.
Response 2: We thank the reviewer for highlighting this critical point, as suggested more technical details were provided in the revised version and repetitions of concepts from the conclusions were eliminated.
Point 3: The main problem is that the Authors focused their attention only on several compounds. Therefore it is not convenient that they stress this point.
Response 3: We thank the reviewer for this highlighting, some repetitions were removed.
Point 4: Other point is that they did not declare how much their method is more sensitive than the other, they only describe the analytical procedures.
Response 4: We would like to thank the reviewer for highlighting this point. A comparison of the sensitivity of the proposed method to others reported in the literature was provided in the revised version, and supported by related and updated references.
Point 5: About the analytical procedures they declare that were conducted by using:
Line 126 “in-house validation”;
Response 5: We thank the reviewer for the correction. We have replaced "in-house validation" with “validation”.
Point6: That is not scientifically acceptable and the major point is about the use of an internal standard in order to normalize the instrumental error procedures witch depends from the instrumental noise that could be change during the time needed to perform all the experiments.
Response 6: We would like to thank the reviewer for this correction. Clarification regarding the calculation of the instrumental noise has been provided in the revised version of the manuscript.
Thanking you in advance for your kind attention,
Sincerely,
Tullia Gallina Toschi
Reviewer 3 Report
Some changes should be done in the manuscript as follows:

Author Response
Response to Reviewer 3 Comments
Point 1: For consistency, and better understanding material and methods section should be explained before the results section. Furthermore, it resembles more a technical report thanan article. E.g. explain extraction (section 3.3) or sampling (sect. 3.2) before the method development (sect. 2.1).
Response 1: We would like to thank the reviewer for this suggestion but we could not change the order of the sections because we followed the template provided by the journal in which the results and discussion section are reported before the materials and methods.
Point 2: L.97 Mobile phases were tested…How many? which? it could be interesting shows this results in a supplementary information file.
Response 2: We thank the reviewer for the very useful suggestion. The preliminary tests carried out for the definition of the analytical conditions in an additional information file are now provided in an additional information file. It can be useful to see the path we followed.
Point 3: L.125 how many times was diluted or concentrated the extract?
Response 3: We thank the reviewer for this question. The dilution values have been better explained in the revised version of the manuscript. Furthermore, the detailed procedure is presented in supplementary material file S1 (Standard operating procedure of the method presented in this article).
Point 4: Section 2.2, in my opinion this section should belong to methodology and not to results section.
Response 4: We thank the reviewer for the suggestion. In the revised version we moved a more theoretical part of this section into the materials and methods, leaving comments on each validation parameter in the results and discussion section.
Point 5: RESULTS only appear in section 3.2 and their discussion should be improved.
Response 5: We thank the reviewer for highlighting this critical point; and in the revised version more detail in the discussion of the results was provided.
Point 6: Table 2, please improve and clarify indicating CV% and % values in the corresponding rows instead of columns (to better understanding).
Response 6: We thank the reviewer for the suggestion; the table has been modified in the revised version.
Point 7: Table 1. The recovery percentage of CBD is 98.8 but in line 162 is indicated a value of 84.92. Please clarify it!
Response 7: We would thank the reviewer for pointing out this error, and the percentage of CBD recovery has been corrected in the Table 1.
Point 8: Line 299-303 Discussion o conclusion?
Response 8: We thank the reviewer for the suggestion. The conclusions have been changed and this sentence has been deleted while that concept was previously explained in the introduction section.
Point 9: abbreviations in abstract should be avoided.
Response 9: We thank the reviewer for this correction; the names of the cannabinoids reported in the abstract for the first time have been written in full and, only later, have been reported with the relative abbreviations, thus respecting the maximum limit ofwords provided for the abstract.
Point 10: sativa must be in lowercase. Please correct it through the MS.
Response 10:We thank the reviewer for highlighting the mistake, “Sativa” was corrected with “sativa” in the revised version.
Point 11: Line 57. Please check the citation reference (8).
Response 11:We would like to thank the reviewer for pointing out this formatting mistake, “8” was corrected with “[8]” in the revised version.
Point 12: Line 135. Standard deviation use to be abbreviated as SD. I suggest changing Sr abbreviation.
Response 12:As suggested, the abbreviations of “standard deviation” was replaced with SD.
Point 13: lines 308 and 309 “calibration curves term in repeated in both sentences” Please correct these mistake.
Response 13:We thank the reviewer for pointing out the typographical error, which has been corrected in the revised version.
Thanking you in advance for your kind attention,
Sincerely,
Tullia Gallina Toschi
Round 2
Reviewer 2 Report
The Author has given proper answers to the previous submitted suggestions.
Moreover several point have to consider in the revised form before the final decision:
· line 85: "in-house validation" change this declaration;
· line 152: "Quantities of CBD (4, 8, and 24 microg / mL)" change with micro with the right font;
· line 176-177: "also in addition, the other cannabinoids (xx, xx, xx) showed low LOQs" please change the (xx, xx, xx) they don't have any mean;
· 289-292: "Precision was calculated with the following formula: CV%= [(SD/x ̅) × 100] 290 where: Sr is the estimate of the standard deviation; x ̅ is the average of the replications made"
please control this exploitation SD and Sr definition
Author Response
Response to Reviewer 2 Comments
Point 1:Line 85: "in-house validation" change this declaration;
Response 1: We thank the reviewer for the suggestion, we changed this declaration.
Point 2: Line 152: "Quantities of CBD (4, 8, and 24 microg / mL)" change with micro with the right font;
Response 2: We would like to thank the reviewer for highlighting this error, we changed from “microg” to “μg”
Point 3: Line 176-177: "also in addition, the other cannabinoids (xx, xx, xx) showed low LOQs" please change the (xx, xx, xx) they don't have any mean;
Response 3: We thank the reviewer for pointing out this oversight, we wrote the name of each cannabinoid instead of “(xx, xx, xx)”.
Point 4: 289-292: "Precision was calculated with the following formula: CV%= [(SD/x ̅) × 100] 290 where: Sr is the estimate of the standard deviation; x ̅ is the average of the replications made"
please control this exploitation SD and Sr definition
Response 4: We would like to thank the reviewer for this suggestion, we controlled the exploitation of SD and Sr. The reported formula refers to the CV% so SD is the correct abbreviation, being a standard deviation. Therefore, we thank the reviewer for pointing out this error, which was corrected in the revised version.
Thanking you in advance for your kind attention,
Sincerely,
Tullia Gallina Toschi